# The Impact on the Therapeutic Decision of Massive Gene Sequencing (NGS) in Plasma from Patients with Advanced Non-Small Cell Lung Cancer (NSCLC)

**DOI:** 10.3390/cancers17213469

**Published:** 2025-10-29

**Authors:** Paula Llor-Rodriguez, Ana Blasco-Cordellat, Sonia Macia-Escalante, Leonor Fernández-Murga, José Vidal-Martinez, Javier Garde-Noguera, José García-Sánchez, Antonio Llombart-Cussac

**Affiliations:** 1Medical Oncology Department, Hospital Arnau de Vilanova-Lliria, Fundación para el fomento de la Investigación Sanitaria y Biomédica de la Comunidad Valenciana (FISABIO), 46015 Valencia, Spain; llor_pau@gva.es (P.L.-R.); garcia_jossane@gva.es (J.G.-S.); llombart@maj3.health (A.L.-C.); 2Medical Oncology Department, Hospital Ascires, 46014 Valencia, Spain; 3SMED Clinical-Research, 03540 Alicante, Spain; 4Clinical and Molecular Oncology Department, Hospital Arnau de Vilanova-Lliria, Fundación para el fomento de la Investigación Sanitaria y Biomédica de la Comunidad Valenciana (FISABIO), 46015 Valencia, Spain; malefer@uv.es; 5Clinical Analysis Department, Hospital Arnau de Vilanova-Lliria, Fundación para el fomento de la Investigación Sanitaria y Biomédica de la Comunidad Valenciana (FISABIO), 46015 Valencia, Spain; vidal_josmar@gva.es; 6Medical Oncology Department, Fundación Instituto Valenciano de Oncología (FIVO), 46009 Valencia, Spain; 7Departamento de Medicina, Facultad de Ciencias de la Salud, Universidad Cardenal Herrera-CEU, Alfara del Patriarca, 46115 Valencia, Spain

**Keywords:** lung cancer, liquid biopsy, biomarkers, personalized medicine, cell-free DNA, circulating tumor DNA

## Abstract

Lung cancer is one of the most frequent and lethal malignancies worldwide, with most cases diagnosed at an advanced stage. Identifying specific genetic alterations is crucial to guide targeted therapies, but obtaining tissue samples is often challenging in routine practice. Liquid biopsy using next-generation sequencing (NGS) enables the detection of tumor-derived DNA circulating in the blood, allowing a non-invasive and comprehensive genomic characterization. In this real-world study, we analyzed plasma samples from patients with advanced non-small cell lung cancer to evaluate the clinical utility of plasma-based NGS for therapeutic decision-making. Our findings show that ctDNA profiling can identify patients eligible for targeted therapies and provides complementary information to tissue analysis. This approach supports the integration of liquid biopsy into routine clinical management of lung cancer, with the potential to improve access to personalized treatment and patient outcomes.

## 1. Introduction

Lung cancer still remains the leading cause of cancer-related mortality worldwide, accounting for more than 1.8 million deaths annually [1]. About 85% of all cases are non-small cell lung cancer (NSCLC), and most patients receive a diagnosis at an advanced stage. This makes curative interventions less accessible and emphasizes the necessity of individualized treatment plans [2]. In recent years, advances in molecular diagnostics have transformed the management of NSCLC by enabling personalized treatments tailored to the specific genomic profiles of each patient.

Tissue-based molecular diagnosis using next-generation sequencing (NGS) is considered the gold standard. Obtaining representative tissue samples in NSCLC is often a challenge, due to tumor location, patient frailty, risks associated with invasive procedures, and marked intratumoral heterogeneity [3]. Even if tissue is available, it might not be enough for repeated analyses as the disease progresses or for thorough genomic testing. Furthermore, because metastatic lesions can develop unique molecular changes over time, a localized biopsy might not adequately capture the spatial and temporal complexity of the disease [4].

In this context, liquid biopsy is a minimally invasive technique, which allows us to detect genetic material derived from the tumor, especially circulating tumor DNA (ctDNA), in peripheral blood [5]. ctDNA comprises small fragments of tumor DNA that are released into the circulation as a result of apoptosis and necrosis. ctDNA offers a high-sensitivity way to identify somatic mutations, gene fusions, copy number changes, and other molecular abnormalities relevant to NSCLC when examined using NGS technologies [5,6].

Sensitivity and concordance between plasma and tissue NGS is high, especially in patients with significant tumor burden, although tissue NGS remains preferred when available. However, NGS in plasma detects additional alterations in a significant percentage of cases and shortens the time to diagnosis and treatment, which has a positive impact on survival and the optimization of resources [7,8,9].

The main guidelines (ESMO, NCCN, ASCO) have included it into diagnostic algorithms to identify target mutations in EGFR, ALK, ROS1, BRAF, MET, RET, NTRK, and KRAS [10,11,12].

Beyond its diagnostic role, liquid biopsy offers valuable advantages for the dynamic monitoring of NSCLC patients. Serial blood sampling facilitates real-time assessment of treatment response and early detection of emerging resistance mechanisms without requiring invasive procedures [13,14]. Longitudinal ctDNA analysis has shown promise not only in tracking therapeutic efficacy but also in detecting minimal residual disease (MRD) and predicting relapse earlier than conventional imaging [15,16]. This capacity is especially critical in NSCLC, where rapid tumor evolution may necessitate timely adjustments to treatment [17].

Despite its potential, liquid biopsy is not routinely performed in clinical practice. Challenges include variability in ctDNA shedding, limited sensitivity for some alterations, and the need for further validation of its impact on clinical outcomes such as progression-free survival (PFS) and overall survival (OS) [18]. Additionally, concordance between tissue and plasma genotyping, while promising, may vary depending on mutation type and tumor burden [17]. For instance, our local data showed a moderate concordance for EGFR mutations between plasma and tissue (57.0%) but a lower detection rate for BRAF and KRAS mutations in plasma [19].

The present study aims to evaluate the clinical utility of plasma-based NGS using the a ctDNA expanded kit (AVENIO) in patients with advanced NSCLC treated in a real-world setting. This kit, based on an NGS platform validated in multicentric studies, provides a means to perform comprehensive genomic profiling from a simple plasma sample [20]. This system can detect single nucleotide variants, insertions and deletions, gene rearrangements, and copy number variations in a preselected panel of clinically relevant genes. The high concordance observed between ctDNA-derived profiles and tissue genotyping, when samples are collected within a short time interval, supports its implementation in clinical management [21,22,23].

The primary objective of our study is to determine whether ctDNA analysis can reliably identify genetic alterations that enable access to targeted therapies or clinical trials, thereby supporting clinical decision-making. In addition, we assess the rate of access to these therapeutic options and the concordance with tissue genetic profiles when available. Through this research, we aim to provide solid and practical evidence to support the integration of liquid biopsy into NSCLC treatment algorithms, with the hope of improving outcomes and the quality of life of our patients.

## 2. Materials and Methods

### 2.1. Study Design and Patients

This was a retrospective observational study conducted at Hospital Arnau de Vilanova in Valencia, Spain. Patients diagnosed with non-small cell lung cancer (NSCLC) by histopathological analysis of tissue samples were included. Plasma samples for circulating tumor DNA (ctDNA) analysis by next-generation sequencing (NGS) were collected between November 2019 and October 2020, either at initial diagnosis or at disease progression.

Informed consent was obtained from each patient. The present study comprised 78 patients aged 18 years and over, diagnosed with stage IV non-small cell lung cancer histopathologically. The clinical staging of the disease was defined in accordance with the guidelines of the tumor-node-metastasis (TNM) staging system in its eighth edition.

The demographic and clinicopathological characteristics, in addition to previous oncological treatments (if applicable), were retrieved from medical records in the hospital informatics system.

### 2.2. Tissue Samples

We retrospectively recorded the results of tissue samples for each patient. Tissue samples were acquired as formalin-fixed and paraffin-embedded tissue. Genomic alterations were evaluated by standard tests from routine clinical procedures, including EGFR polymerase chain reaction; ALK IHC or fluorescence in situ hybridization (FISH); KRAS polymerase chain reaction; ROS1 immunohistochemistry, FISH; or reverse transcriptase-polymerase chain reaction.

### 2.3. Next-Generation Sequencing (NGS)-Based Liquid Biopsy

Approximately 4 mL of a blood sample was collected from each patient at the time of diagnoses or at progression of disease. Extraction of ctDNA and NGS-based liquid biopsy was performed by Avenio Expanded Kit. Avenio Expanded Kit detects ct-DNA in blood specimens and evaluates exons from 77 genes, reporting point mutations (as single-nucleotide variants, SNVs), insertion/deletions, copy number amplifications, and fusions/rearrangements, offering researchers a detailed insight into genetic alterations.

### 2.4. Outcomes and Statistical Analyses

Primary outcomes were (i) the prevalence and spectrum of pathogenic/likely pathogenic alterations detected by plasma NGS; (ii) concordance between plasma-based and tissue-based genotyping for predefined drivers (EGFR, KRAS, BRAF, MET, etc.); and (iii) overall survival (OS) in relation to receipt of NGS-guided therapy. OS was defined from the date of liquid biopsy to death from any cause or last follow-up (data cutoff: 1 February 2021).

Descriptive statistics summarized baseline features, molecular findings, and treatments (continuous variables as median [IQR]; categorical as n [%]). Group comparisons used the Wilcoxon rank-sum test for continuous variables and Fisher’s exact test (or χ^2^ when appropriate) for categorical variables; two-sided α = 0.05.

### 2.5. Ethical Considerations

This study was approved by the Ethics Committee of Hospital Arnau de Vilanova de Valencia (Reference: HAV-BAR-2020-03). Data confidentiality was maintained in accordance with current Spanish legislation (Organic Law 03/2018 on Data Protection, published in BOE no. 294, BOE-A-2018-16673).

The ethics committee waived the requirement for informed consent in cases where the patient had died or where clinical conditions precluded its collection. The authors declare no relevant affiliations or financial interests related to the subject matter of this manuscript.

## 3. Results

### 3.1. Patient Characteristics

Between 1 November 2019 and 31 October 2020, a total of 78 patients were included in this study. The median age was 63.5 years, 52 (66.7%) were male (66.7%), and all of them presented advanced NSCLC at the moment of the inclusion in the study. Histology was adenocarcinoma in 67 cases (85.9%), squamous in 5 (6.4%), and others in 6 cases (7.8%). Twelve patients (15.4%) were non-smokers, thirty-eight former smokers (48.7%), and twenty-six active smokers. ECOG performance status was 0–1 in 63 (80.8%), 2 in 13 (16.7%), and 3 in 2 (2.6%). At the time of liquid biopsy, 58 (74.4%) were treatment-naïve, 15 (19.2%) were on second-line, and 5 (6.4%) on third-line therapy. Clinical and pathological characteristics are summarized in Table 1.

### 3.2. Molecular Alterations Detected by Massive Gene Sequencing (NGS) in Plasma

KRAS mutation was the most frequent pathogenic alteration, accounting for 16.9% of cases, followed by MET (7.8%), PiK3CA and EGFR (both at 6.5%), and ERBB2 and IDH2 (both 6.5%). Additional pathogenic variants were identified in single patients (1/78, 1.3% each) and included TP53, NRAS, BRCA1, BRCA2, FLT3, PTEN, and JAK2. In total, 8/78 (10.37%) patients received ctDNA-NGS-guided treatment: EGFR (5/5, 100%), KRAS (2/13, 15.4%), and BRCA1 (1/1, 100%). No ctDNA-guided treatment was initiated for alterations in PIK3CA, TP53, MET, ERBB2, NRAS, BRCA2, IDH2, FLT3, or PTEN. Among likely pathogenic variants, TP53 was most frequent (12/78, 15.6%), followed by RB1 (3/78, 3.9%) and PTEN, APC, PTCH1, and KEAP1 (each 2/78, 2.6%); FGFR, RET, CTNNB1, IDH1, NFE2L2, MSH2, and RAF1 were each observed in one patient (1/78, 1.3%). No ALK or ROS1 rearrangements were detected. BRAF V600E was rare (3/78, 4.3%); in our cohort, these events were identified in tissue but not confirmed in plasma ctDNA.

### 3.3. Concordance Between Massive Gene Sequencing (NGS) in Plasma and Tissue

Tissue–plasma concordance: Comparison of tissue-based genotyping with plasma NGS showed moderate agreement for EGFR (overall percent agreement, OPA 57.0%), with tissue and plasma positivity rates of 11.4% and 6.5%, respectively. For BRAF V600E, none of the tissue-positive cases were detected in plasma (positive percent agreement, 0%), consistent with limited ctDNA shedding and/or assay sensitivity. KRAS exhibited substantial discordance, with a higher detection rate in plasma than in tissue (16.9% vs. 2.6%). These findings underscore the complementary roles of tissue and plasma testing and the importance of synchronized sampling (Figure 1).

### 3.4. Survival Outcome in Patients Regarding Treatment Guided by Liquid Biopsy

On unadjusted Kaplan–Meier analysis, overall survival (OS) was numerically longer in patients who received NGS-guided therapy than in those who did not; however, the difference was not statistically significant (log-rank *p* = 0.344) (Figure 2).

## 4. Discussion

Non-small cell lung cancer (NSCLC) is the leading cause of cancer-related mortality worldwide [1]. The advent of targeted therapies has radically transformed its management, placing precise identification of actionable genomic alterations at the core of personalized treatment planning. The implementation of next-generation sequencing (NGS) techniques in liquid biopsy represents a major advance in the molecular diagnosis and longitudinal monitoring of NSCLC. Our study evaluates the clinical impact of plasma ctDNA sequencing in a real-world clinical setting. Our findings reinforce the value of ctDNA as a complementary or alternative tool to tissue biopsy, both in initial therapeutic decision-making and in understanding the dynamic evolution of the disease.

A key finding in our analysis is that 10.25% of patients accessed targeted therapy based exclusively on ctDNA molecular findings. Although modest, this percentage is especially relevant given that, in routine clinical practice, access to tumor tissue may be limited by tumor location, the patient’s clinical condition, or the urgency to initiate treatment. In our study, more than 60% of cases were analyzed in the first-line setting, a critical moment for maximizing therapeutic benefit through early initiation of targeted therapy. Our data are consistent with larger prospective studies indicating that early use of plasma NGS facilitates access to targeted therapies, even in the absence of available tumor tissue or when tissue does not adequately reflect the dominant tumor clone [21,23].

In our cohort, the most frequently detected mutations in plasma were KRAS, MET, EGFR, and PIK3CA, matching the mutational profiles reported in reference series such as those by Zill [22] and Fernández-Murga [19]. However, as also observed in the study by Andrews et al. [24], not all ctDNA-detected alterations lead to an immediate change in therapy, though they still represent an opportunity for individualized intervention.

Comparison of molecular findings between tissue and ctDNA revealed a moderate concordance for EGFR (57%), which, although lower than in controlled clinical trials, is representative of real-world practice. This is in line with data from multicenter studies [19,20,21,22,23,24]. The lower concordance may be explained by the time interval between sample collections, variable tissue sample quality, and the intrinsic intratumoral heterogeneity. Recent publications [24] emphasize the importance of synchronized sample collection to improve concordance and point out that ctDNA detection may be limited in patients with brain-only disease or low tumor burden, underscoring the need for cautious interpretation of negative plasma results.

In our series, concordance was particularly low for other alterations such as BRAF and KRAS, with lower detection rates in plasma, similar to what has been reported in other cohorts [19,20,21,22,23]. This suggests that some molecular alterations may be underrepresented in ctDNA, or that tumor DNA shedding into the circulation varies by molecular subtype. Therefore, combining both tissue and plasma analyses provides a more comprehensive view of the patient’s molecular profile and highlights the need for clear integration criteria in therapeutic algorithms [4,9].

Another crucial aspect is the utility of ctDNA for real-time monitoring of disease evolution. While our study did not include serial ctDNA analyses, recent research has demonstrated its usefulness for detecting minimal residual disease (MRD) [15,16], as well as for the early identification of acquired resistance mechanisms and even detection of progression weeks or months before radiologic confirmation [24]. The concept of molecular progression reinforces the role of liquid biopsy not only as a diagnostic tool but also as a dynamic biomarker of tumor evolution, enabling earlier and adaptive intervention—an approach that warrants validation in prospective clinical trials [25,26].

From a technical perspective, the AVENIO platform demonstrated analytical robustness, broad genomic coverage, and high sensitivity and specificity. Platform selection should be aligned with the quality standards set by organizations such as IASLC or ESMO to ensure reproducibility and clinical relevance of the results obtained in different studies [10,13].

Regarding prognostic impact, our data show a favorable trend in overall survival for patients who received targeted therapies guided by ctDNA results. Although this difference did not reach statistical significance, it is consistent with the clinical benefit observed in larger cohorts and supports the hypothesis that early identification of relevant molecular alterations can influence the disease course. Our study generates a hypothesis to be tested in future, adequately powered research.

Our work has several limitations: the retrospective design, limited sample size, lack of serial ctDNA analyses, and heterogeneity in lines of therapy may limit the interpretation of results. In addition, tissue samples were not always available for comparison, which constrained concordance analysis. Nevertheless, our study demonstrates the feasibility of incorporating liquid biopsy into routine oncology practice. Its consistency with prospective data supports its validity and underscores the need for systematic integration of liquid biopsy into clinical workflows. The future of oncology lies in integrative models that combine tissue- and plasma-derived information with clinical, radiologic, and even artificial intelligence-based data for personalized prognostic and therapeutic stratification. In this evolving scenario, ctDNA is an essential tool.

## 5. Conclusions

In conclusion, our study demonstrates that plasma-based next-generation sequencing (NGS) is a feasible, minimally invasive, and clinically informative tool for the molecular profiling of patients with advanced non-small cell lung cancer. ctDNA analysis not only identifies actionable genomic alterations but also provides complementary information to tissue-based testing, particularly in scenarios where tumor biopsy is not feasible or representative. The integration of liquid biopsy into clinical decision-making has the potential to enhance patient access to targeted therapies, enable real-time disease monitoring, and support personalized treatment strategies. While further prospective studies are warranted to validate its prognostic and predictive value, our findings support the implementation of plasma NGS as part of routine molecular diagnostics in advanced NSCLC.

## Figures and Tables

**Figure 1 cancers-17-03469-f001:**
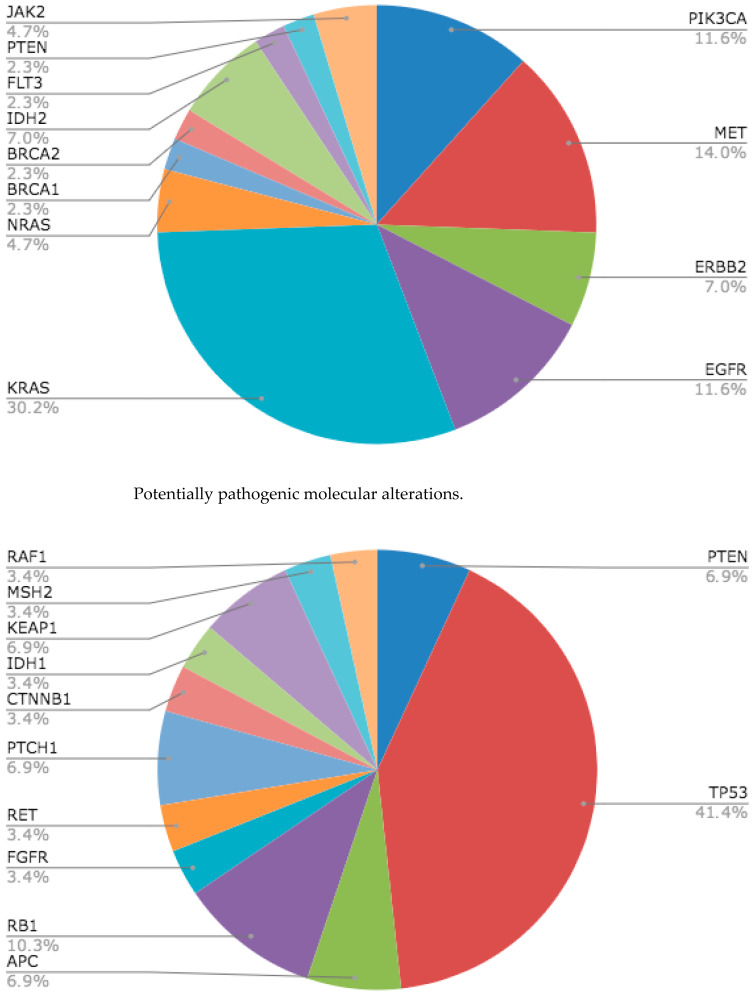
Pathogenic molecular alterations.

**Figure 2 cancers-17-03469-f002:**
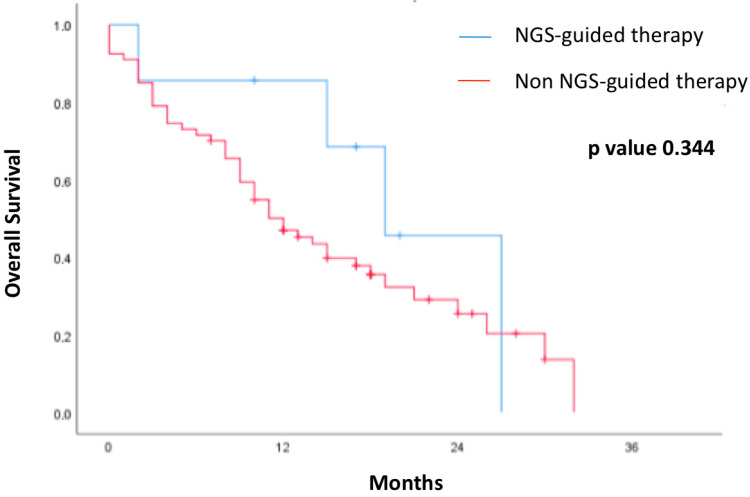
Overall survival according to NGS-based therapy.

**Table 1 cancers-17-03469-t001:** Patient characteristics.

Characteristics	N (%)
**Total Patients**	78
Male	52 (66.7)
Female	26 (33.3)
**Age (in years)**	
Median	63.5
Range	29–90
**Histology**	
Adenocarcinoma	67 (85.9)
Squamous carcinoma	5 (6.4)
Other	6 (7.8)
**Smoking Status**	
Non-smoker	12 (15.4)
Former smoker	38 (48.7)
Active smoker	26 (33.3)
**ECOG PS**	
0–1	63 (80.8)
2	13 (16.7)
3	2 (2.6)
**Previous Treatment**	
0	58 (65.4)
1	15 (19.2)
≥2	5 (6.4)

## Data Availability

The original contributions presented in this study are included in the article. Further inquiries can be directed to the corresponding author.

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
