# Peer review of "The Impact on the Therapeutic Decision of Massive Gene Sequencing (NGS) in Plasma from Patients with Advanced Non-Small Cell Lung Cancer (NSCLC)"

_cancers, 2025, doi:10.3390/cancers17213469_

Round 1
Reviewer 1 Report
Comments and Suggestions for Authors
1. Line 45: Authors can still cite the most recent Cancer statistics article instead of 2021 article.
2. Line 125: Authors should cite article for Avenio Expanded Kit
3. In the methods section, the authors should also mention how the data was analyzed
4. The conclusions are made from a panel based NGS kit, as opposed to whole genome sequencing, which makes the conclusions focused only on the 77 known oncogenes.
5. Authors mention that patients were then given NGS-guided therapy. Here it is crucial to state in the main text or in the supplementary data what kind of therapy was recommended, was it a small molecule based therapy?
Author Response
- Line 45: Authors can still cite the most recent Cancer statistics article instead of 2021 article.
- Line 125: Authors should cite article for Avenio Expanded Kit
- In the methods section, the authors should also mention how the data was analyzed
- The conclusions are made from a panel based NGS kit, as opposed to whole genome sequencing, which makes the conclusions focused only on the 77 known oncogenes.
- Authors mention that patients were then given NGS-guided therapy. Here it is crucial to state in the main text or in the supplementary data what kind of therapy was recommended, was it a small molecule based therapy?
- We have added more recent article of cancer-statistics.
- We have added the cite for Avenio Expanded Kit
- In the Materials and Methods section, the statistical methodology applied for the analysis is described, which is primarily descriptive.
- Indeed, the analysis is limited to the genes included in the panel; however, this panel comprises all those that can potentially be targets for targeted therapies.
- We have added the information regarding patients treated with targeted therapy and clarified that all of them were indeed treated with oral small-molecule–based therapies.
Reviewer 2 Report
Comments and Suggestions for Authors
Dear authors,
Your article is very interesting: ctDNA is a promising technique to search molecular target and may also have a role in monitoring disease, but current its use in clinical practice is not yet well defined.
I've only some suggestion: first of all, can you explain what is your standard for testing patients (tissue NGS or PCR?), because may be interesting the rate of concordance between tissue and liquid NGS, and also the rate between other test (such as PCR) and ctDNA: this may underscore the value of ctDNA in clinical practice.
Another information that may be useful is, for patients tested in second or greater line and found to have a genomic alteration, if they received a target therapy in previous line: may the treatment receveid have an impact on circulating DNA.
Another info I would suggest to add is the turn around time, if avaible: you know that this is a critical point, especially in second or greater lines of treatment.
Last, I would add the info if all the patients receveid treatment as clinical practice or someone received it in a clinical trial: this may have an impact on overall survival.
Author Response
Your article is very interesting: ctDNA is a promising technique to search molecular target and may also have a role in monitoring disease, but current its use in clinical practice is not yet well defined.
I've only some suggestion: first of all, can you explain what is your standard for testing patients (tissue NGS or PCR?), because may be interesting the rate of concordance between tissue and liquid NGS, and also the rate between other test (such as PCR) and ctDNA: this may underscore the value of ctDNA in clinical practice.
Another information that may be useful is, for patients tested in second or greater line and found to have a genomic alteration, if they received a target therapy in previous line: may the treatment receveid have an impact on circulating DNA.
Another info I would suggest to add is the turn around time, if avaible: you know that this is a critical point, especially in second or greater lines of treatment.
Last, I would add the info if all the patients receveid treatment as clinical practice or someone received it in a clinical trial: this may have an impact on overall survival.
Thank you very much for your comments; these are very interesting points for the discussion.
At the time the study was initiated, liquid biopsy was often reserved for patients in whom PCR testing for the most frequent genes (EGFR, KRAS, and BRAF) was negative. This explains the low detection rate of these genes in our series. Moreover, in a significant proportion of patients, the analysis was performed in the second line or later, which can indeed affect the sensitivity of the technique for ctDNA detection.
Unfortunately, we did not collect the turn-around time, and therefore we are unable to provide this information.
All these patients where treated as clinical practice, none of them was included in clinical trial
We take in consideration your comments and add this information to the manuscript.